

# A majority voting framework for reliable sentiment analysis of product reviews

Darie Moldovan[1,2]

[1] Business Information Systems, Babes-Bolyai University of Cluj-Napoca, Cluj-Napoca, Romania
[2] Rutgers, The State University of New Jersey-Newark, Newark, NJ, United States

## ABSTRACT

This article presents a tailored majority voting approach for enhancing the consistency and reliability of sentiment analysis in online product reviews. The methodology addresses discrepancies in sentiment classification by leveraging sentiment labels from multiple automated tools and implementing a robust majority decision rule. This consensus-based approach significantly enhances the trustworthiness and consistency of sentiment analysis outcomes, serving as a dependable foundation for training more precise sentiment analysis models. The data labeled with our method was utilized to train deep learning models, achieving competitive accuracy with significantly less data. The findings demonstrate the effectiveness of the method in producing results comparable to commercial tools while ensuring data consistency for model training.

## INTRODUCTION

Businesses increasingly rely on sentiment analysis to understand customer preferences, perform market analysis, monitor brand reputation, and gauge customer experiences, making it a powerful tool for decision-making in competitive markets (*Geetha & Karthika Renuka, 2021*). While originally intended as a feedback tool to help companies improve, product reviews now also influence consumer perception and drive sales rankings (*Catelli et al., 2022*; *Kim, 2024*). However, the reliability of online opinions remains questionable, as user-generated content often lacks regulation, leading to issues like spam, irrelevant posts, and fake reviews. Compounding these issues, the ground truth for determining whether an opinion is positive, negative, or neutral is frequently unavailable, complicating the development of accurate sentiment classifiers (*Fang & Zhan, 2015*; *Hassan & Islam, 2021*; *Sadiq et al., 2021*; *Liu et al., 2023*).

User ratings pose additional challenges. These ratings are inherently subjective, influenced by personal preferences, biases, and expectations, which can result in discrepancies between a rating and the textual sentiment in the review. For instance, two users may give identical ratings while expressing very different sentiments in their written feedback, creating ambiguity in sentiment interpretation. Additionally, ratings may be used for purposes beyond expressing sentiment—such as signaling satisfaction on specific attributes or influencing others' purchase decisions. Without understanding the intent

Corresponding author
Darie Moldovan,
darie.moldovan@econ.ubbcluj.ro

behind these ratings, inferring sentiment solely from numerical values becomes difficult and potentially misleading (*Han & Anderson, 2020*).

Moreover, sentiment expression may vary across product categories, meaning that ratings alone cannot reliably generalize sentiment labels across diverse datasets (*Mao, Liu & Zhang, 2024*). These challenges underscore the need for high-quality labeling that accurately reflects sentiment, especially when training models that rely on clearly defined examples to learn effectively. Sentiment labeling is typically performed by humans to capture nuances in language, but for massive datasets, this manual approach is impractical, time-consuming, and cost-prohibitive (*Shayaa et al., 2018*; *Van Atteveldt, Van der Velden & Boukes, 2021*); machine-assisted labeling, while helpful, may introduce inconsistencies that complicate model training (*Van Atteveldt, Van der Velden & Boukes, 2021*; *Wankhade, Rao & Kulkarni, 2022*).

When training a sentiment analysis model, avoiding ambiguity in labeling the dataset is crucial. Ambiguous or inconsistent labeling of sentiment can lead to noise in the dataset, which can adversely affect the performance and accuracy of the trained model (*Biswas, Young & Griffith, 2022*). Ambiguous labels may introduce confusion and uncertainty into the training process, making it difficult for the model to learn and generalize patterns effectively.

Moreover, ambiguous sentiment labels can result in biased or skewed training data, leading to biased predictions and inaccurate sentiment analysis outcomes. This can have significant implications, especially in applications where precise sentiment analysis is critical, such as customer feedback analysis, market research, and sentiment-based decision-making processes. Additionally, ambiguous sentiment labels can hinder the interpretability and trustworthiness of the sentiment analysis results. Stakeholders relying on the output of sentiment analysis models may struggle to interpret or trust the results if they are influenced by ambiguous or inconsistent labeling practices (*Esposito, Moscato & Sperlí, 2021*).

Our research addresses the need for reliable and accurate sentiment analysis by introducing the majority voting method for labeling sentiment on reviews. By incorporating this method into sentiment analysis, we aim to mitigate the challenges associated with ambiguous sentiment labeling and improve the trustworthiness of sentiment analysis outcomes.

The majority voting method offers a systematic and transparent mechanism for resolving discrepancies in sentiment classification, particularly in cases where multiple sentiment analysis tools produce divergent results. By aggregating sentiment labels from multiple sources and applying a majoritarian decision rule, we create a consensus-based approach that enhances the robustness and reliability of sentiment analysis. When evaluating manually annotated labeling, Krippendorff's alpha (*Krippendorff, 2018*) serves as a metric to measure the consensus among evaluators or voters, thereby indicating the reliability of the results. We apply the same method to assess the consensus among automated labeling agents and to determine the incremental value of our proposed approach.

The sentiment classifiers integrated into the majority voting system were selected based on their diversity and complementary methodologies, including machine learning, linguistic analysis, and rule-based approaches. Tools such as Google Cloud Natural Language API, Amazon Comprehend, IBM Watson NLU, and Azure AI Text Analytics were chosen as commercial products from market leaders. First, we assessed the alignment of their responses when evaluating sentiment. Additionally, we sought to leverage their diversity to ensure robustness by combining the strengths of individual classifiers while mitigating their limitations. The approach creates a consistent pseudo-ground truth, improving the reliability and effectiveness of the sentiment analysis process.

By leveraging the majority voting among automated agents, we ensure consistency and accuracy in sentiment annotation, thereby enhancing the quality and integrity of the training data. We apply the methodology to label sentiment for three different-sized datasets of reviews and then train four architectures of deep learning methods to perform sentiment analysis on new data. Recurrent neural networks (RNN), gated recurrent units (GRU), long-short term memory (LSTM) and bidirectional encoder representations from transformers (BERT) architectures were chosen to perform this task.

Additionally, we conduct extensive experiments to assess the impact of dataset size on model convergence and correlation with reference models, providing valuable insights into the scalability and robustness of our proposed methodology. We conduct the tests on a larger, separate dataset, comparing the results with the output of the reference methods.

The remainder of the article is structured as follows. We commence with an up-to-date literature review on sentiment analysis research practices and the efficacy of deep learning in training sentiment analysis models. Next, we elaborate on our proposed methodology, delineating all requisite steps. Subsequently, we present the data and describe the experimental setup. The results are detailed for each step in our methodology, and we discuss the findings. Finally, we conclude with insights and perspectives for future work.

# RELATED WORK

## On the importance of sentiment analysis

A commonly cited marketing adage asserts that "there is no such thing as bad publicity." However, as noted by *Berger, Sorensen & Rasmussen (2010)*, the outcome is significantly influenced by various factors. A negative review for a well-known brand could potentially decrease sales, whereas a similar review for an unfamiliar product might act as a catalyst. In contemporary times, electronic word of mouth has emerged as a primary source of information for consumers of products or services, and its impact is indisputable. *Babić Rosario, De Valck & Sotgiu (2020)* comprehensively categorize its diverse forms in the literature (*e.g.*, sentiment, consumer knowledge, user-generated content), depending on the source of interest. While the vast majority of the shoppers consult the reviews before purchasing a product or a service, the positive rating is definitely an uplift for the buying probability. However, there is no linear relationship between ratings and the probability of making a purchase. A perfect score can raise suspicions among customers, leading them to be tempted to buy products or services with slightly less favorable reviews. Moreover, a

price discount for products with low ratings will be seen by potential clients as a sign of low quality, in the end hearting sales (*Maslowska, Malthouse & Bernritter, 2017*; *Kim, 2024*).

Acknowledging the importance of reviews for the propensity to buy, the phenomenon of reviews manipulation started to become relevant. Although beneficial for merchants in the short term, manipulation leads to a lower confidence in the relevance of reviews and affects the perceived value of the product or service in the long term (*Li et al., 2021*; *Wang et al., 2023*; *Kim, 2024*).

Furthermore, understanding sentiments in the marketing process is essential, as research indicates that individuals can experience emotions triggered by events affecting others. This phenomenon, known as emotional contagion, allows people to feel the same emotions as those they observe. According to *Hossain & Rahman (2023)*, emotion is defined as a positive or negative experience linked to a specific pattern of physiological activity. To determine an individual's empathy, an objective assessment by an external observer or a physiological evaluation would be preferable. In practice, however, empathy is typically measured through questionnaires or narrative situations where participants indicate to what extent they identify or agree with specific statements or stories.

For an organization, it may no longer be necessary to conduct opinion surveys and focus groups to gather public opinions, because there is a wealth of such information available to the public. Social media networks have played a crucial role in reshaping businesses and influencing public sentiments and emotions, resulting in a profound impact on our social and political systems (*Zhang, Wang & Liu, 2018*). Consequently, the collection and study of opinions have become a necessity.

## Methods for sentiment analysis

Sentiment analysis involves the extraction and categorization of emotions embedded in textual data, ranging from basic sentiments like positive, negative, and neutral to more nuanced emotional states such as joy, anger, and sadness. This process is integral in understanding consumer feedback, social media posts, and other user-generated content. Frameworks such as Ekman's six basic emotions (*Ekman, 1992*) provide theoretical foundations for emotion categorization. However, accurately capturing textual emotions presents challenges due to phenomena like sarcasm, cultural context, and negation (*Deng & Ren, 2021*). For instance, a phrase like "Just great!" can express satisfaction or frustration depending on the context.

Supervised learning methods form a core part of traditional sentiment analysis, relying on labeled datasets to train models for sentiment classification. Techniques such as naïve Bayes, support vector machines (SVM), and logistic regression have been widely used due to their robustness and simplicity. For instance, SVMs are particularly effective in defining decision boundaries for binary sentiment classification tasks, while naïve Bayes applies probabilistic techniques to estimate sentiment scores (*Liu, 2022*). Despite their effectiveness, these methods often fail to capture the deeper contextual and syntactic relationships required for nuanced sentiment analysis. Moreover, the reliance on high-quality labeled datasets is a significant limitation, as creating such datasets can be resource-intensive and prone to subjectivity in labeling (*Medhat, Hassan & Korashy, 2014*).

 

Unsupervised learning methods aim to mitigate the dependency on labeled datasets by leveraging algorithms that uncover hidden patterns within textual data. Clustering algorithms like K-means and topic modeling techniques such as latent Dirichlet allocation (LDA) have been applied to group texts by sentiment or extract sentiment-related themes (*Cambria et al., 2013*). Lexicon-based approaches, such as SentiWordNet (*Baccianella, Esuli & Sebastiani, 2010*) or Vader (*Hutto & Gilbert, 2014*), utilize predefined sentiment scores for words to infer the overall sentiment of a text. While these approaches provide valuable insights, they sometimes struggle with context sensitivity and polysemy—issues where the meaning of words changes depending on their usage. These methods were considered in the past to be more suited for exploratory purposes than for producing highly accurate sentiment classifications (*Sharma & Dey, 2012*).

Recent advancements in unsupervised and lexicon-based sentiment analysis have improved their accuracy and contextual understanding, like semantic document representation techniques attempting to enhance fine-grained sentiment analysis by capturing nuanced aspects (*Fu & Cheng, 2019*). Lexicon-based methods have also evolved, addressing context sensitivity and domain-specific limitations (*Barik & Misra, 2024*).

For a broad view of the above mentioned methods see the reviews of *Taboada et al. (2011)*, *Jain, Pamula & Srivastava (2021)*, and *Liu (2022)*.

As in many other domains, deep learning has become a common approach when processing natural language. Different methods expanded rapidly, on the basis of their applicability to specific problems.

*Li, Goh & Jin (2020)* emphasize that while advanced classification algorithms have been developed to improve performance, the textual quality of the data, such as word count and review readability, has often been overlooked. When applying deep learning techniques, their findings suggest that reviews with short length and high readability perform best compared to other combinations of word count and readability levels. They also found that controlling the length of the review is more effective in achieving higher accuracy than increasing readability. *Fang & Zhan (2015)* used a bag-of-words model to compute a score to label reviews as positive or negative, then used ML methods to train a classifier.

Combining a sentiment lexicon and deep learning techniques to analyze e-commerce product reviews in Chinese proved useful (*Yang et al., 2020*). The model utilizes the convolutional neural network (CNN) and attention-based bidirectional GRU to extract sentiment and context features from reviews and then classifies the weighted sentiment features.

*Yang et al. (2020)* emphasize the significance of sentiment analysis in evaluating consumer product reviews on e-commerce platforms, particularly due to issues such as inconsistency between product descriptions and actual goods, poor product quality, and inadequate after-sales services. To highlight the challenge of distinguishing between genuine and misleading reviews, a task that has led to the development of various classification methods, *Catelli et al. (2022)* propose a multi-label classification methodology that utilizes the BERT neural language model to construct a deceptive review detector.

*Kaur & Sharma (2023)* use a deep learning-based model (LSTM) for the analysis of consumer sentiment. In the pre-processing stage, using natural language processing (NLP) techniques, undesirable data from input text reviews are eliminated. A hybrid method is introduced for feature extraction, which includes review-related features and aspect-related features, to construct a unique hybrid feature vector for each review. An attention based LSTM is useful in improving the classification, as it allows the algorithm to focus on the important aspects of the input, an issue difficult to address especially with big data (*Elangovan & Subedha, 2023*).

*Purohit & Patheja (2023)* proposed a novel technique called Revival Extraction, which focuses on extracting specific products based on thematic analysis method to obtain accurate data. They use a so called feedback neural network for combining product aspect feedback loop, and SVMs with bag-of-words for classifying pre-trained review comments with high accuracy.

*Geetha & Karthika Renuka (2021)* explored the use of BERT to improve the performance of aspect-based sentiment analysis. They used various models for comparison, such as naïve Bayes, SVMs, and LSTM. However, they found that many existing sentiment analysis techniques for customer online product review text data have low accuracy and often take a long time during training. The BERT model showed improved performance with good prediction and high accuracy compared to the other machine learning methods. Another aproach based on aspect, and taxonomy was presented by *Tarnowska & Ras (2019)*, targeting the same concerns, like ambiguity in the context of customer reviews. Reducing the sentence length by extracting the relevant content has been identified as a critical step in improving the BERT performance, which can be prone to error where the length of the input is high (*Ansar et al., 2021*).

The literature review (Table 1 provides a summary of the main characteristics of the NLP methods) highlights various NLP techniques used for sentiment analysis, including lexicon-based approaches, machine learning classifiers, deep learning, and hybrid models. Each technique demonstrates unique strengths and limitations, such as the reliance on large annotated datasets for machine learning classifiers or the high computational cost of deep learning models. A key challenge common to all approaches, particularly when applied to e-commerce platforms, is ensuring alignment between rating systems and the content of customer reviews. While prior research has primarily focused on improving the performance of sentiment classifiers or addressing limitations like domain adaptation and computational cost, our study contributes by addressing the concordance issue through a systematic majority voting mechanism. By aggregating the outputs of multiple sentiment analysis tools, refined by the rating labels of the reviews, we provide a robust and practical method for validating and improving classifier reliability.

## METHODOLOGY

This section details the methodology employed for building a sentiment analysis model for online product reviews. The process unfolds in three stages: data pre-processing, sentiment detection for building the training dataset, and model training with deep learning (Fig. 1).

**Table 1 Summary of NLP for sentiment analysis.**

| NLP technique | Feature selection mechanism | Dataset category | Algorithms used | Main shortfall | References |
|---|---|---|---|---|---|
| Lexicon based approach | Manual or Predefined Lexicons | Product reviews, Social media | Naive Bayes, Lexicon based | Performance varies | *Liu & Shen (2020)*, *Barik & Misra (2024)* |
| ML classifiers | Term frequency (TF), Mutual information | Social media posts | Logistic regression, Decision trees, XGBoost | Requires large annotated datasets | *Medhat, Hassan & Korashy (2014)*, *Liu (2022)* |
| Unsupervised learning | Unsupervised feature extraction | Customer reviews | K-means, LDA | Lexicon-based methods may be limited in domain adaptation | *Cambria et al. (2013)*, *Al-Ghuribi, Noah & Tiun (2020)* |
| Deep learning | Word embeddings | News articles | LSTM, CNN | High computational cost | *Zhang, Wang & Liu (2018)*, *Liu & Shen (2020)* |
| Transfer learning | Pre-trained embeddings | Tweets, Product reviews | BERT variants | May require fine-tuning for specific domains | *Tao & Fang (2020)*, *Tan et al. (2022)* |
| Hybrid models | Ensemble learning | Tweets, Reviews, Emails | SVM, LSTM, CNN | Ensemble models can be computationally expensive | *Dang, Moreno-García & De la Prieta (2021)*, *Janjua et al. (2021)* |

Data pre-processing involves standard NLP techniques. First, all text was converted to lowercase to normalize case sensitivity. Next, the text was broken down into individual words, a process called tokenization. After this, common words like "the," "and," and "is," which do not carry significant meaning, were removed. These are known as stop words, and we used the English stopword list from the Natural Language Toolkit (NLTK) (*Bird, Klein & Loper, 2009*) to identify them. Then, the remaining words were simplified to their basic dictionary form through a process called lemmatization. For example, the word "running" might be reduced to "run." Finally, the cleaned-up words were put back together into a single string, ready for further analysis. It's important to note that this pre-processing step is not required for the automatic sentiment classifiers employed later.

Sentiment detection utilizes a two-pronged approach. First, pre-processed text is subjected to sentiment analysis using the VADER lexicon from the NLTK library (*Bird, Klein & Loper, 2009*). VADER assigns a compound sentiment score ranging from −1 (highly negative) to +1 (highly positive).

Second, an ensemble sentiment classification approach is leveraged. Unlike the pre-processed reviews fed to VADER, the original reviews are directly processed by several pre-trained automatic sentiment classifiers including Google Cloud Natural Language API, Amazon Comprehend, IBM Watson Natural Language Understanding, and Azure AI Text Analytics. Each classifier independently assigns a sentiment label (positive, negative, or neutral/mixed) to the review.

Following sentiment detection, a crucial step in our methodology is sentiment label fusion, which combines the VADER score and the labels from the ensemble classifiers for each review. This is a part of our proposed majority voting mechanism with rating

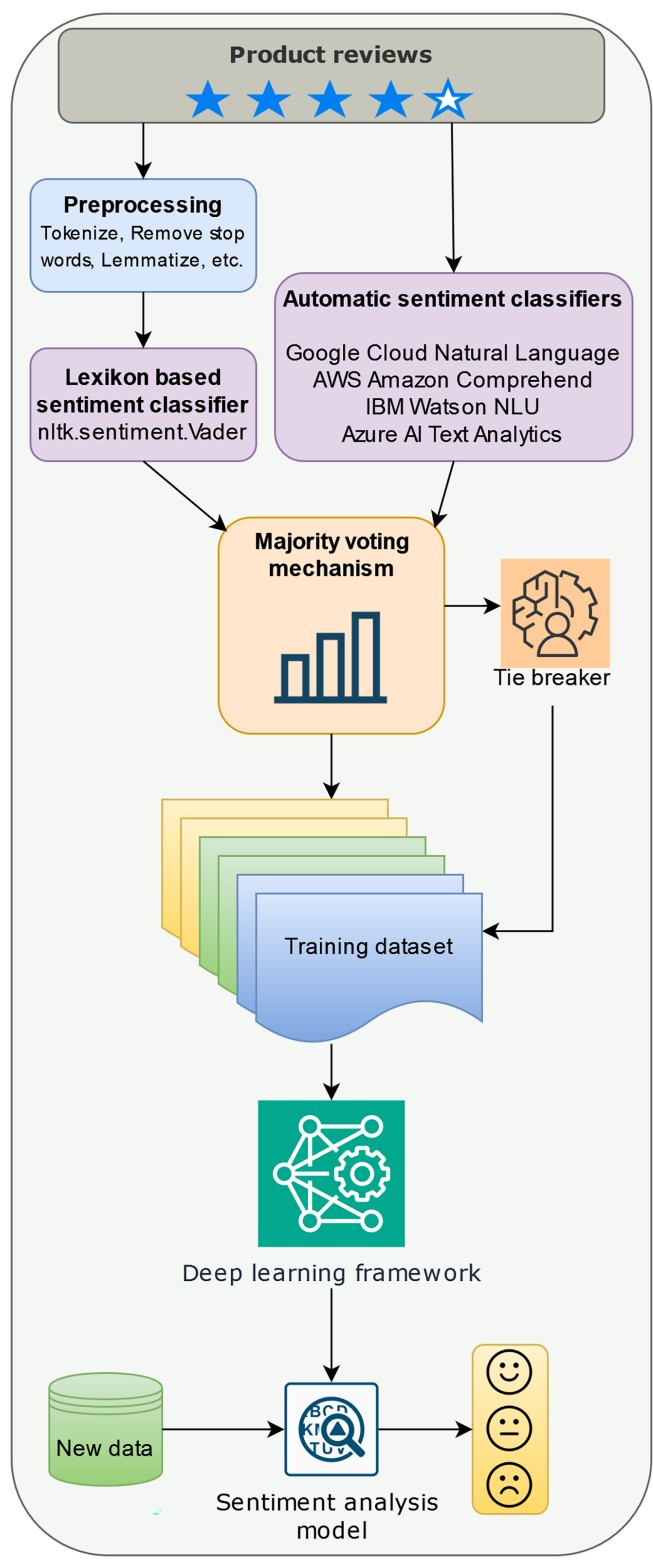

**Figure 1 Building a sentiment analysis model with a majority voting labeling mechanism.**

integration. In cases where the ensemble classifiers provide conflicting labels (equal positive, negative, and neutral/mixed votes), the review's original star rating (on a scale of 1 to 5) is considered.

For instance, consider a review titled "Value for money" with the following content:

*Value for money. Pros 1. Cooling effect—gives better cooling 2. Air delivery—strong and powerful fan 3. Capacity—88 litres Cons. 1. Noise—Fan noise and water dripping noise 2. Material used—Looks like cheap plastic in some corners. 3. Power cable—too small, really need an extension to use. Only for people who wants to sleep in silent place—Don't buy this.*

This review offers both positive aspects (*e.g.*, cooling efficiency and air delivery) and negative ones (*e.g.*, noise and material quality). However, sentiment classifications from various tools diverge significantly: IBM and Microsoft tools label it as negative, VADER assigns it a positive sentiment score, and Amazon and Google classify it as neutral/mixed. The user rating for the review, however, is a 4, suggesting an overall positive sentiment. This inconsistency highlights the limitations of automated sentiment classifiers in complex reviews. Despite the majority voting mechanism resolving many such conflicts, this specific example results in a tie, as neither the positive rating (4) aligns with the classifications of negative or neutral/mixed. In such scenarios, human intervention becomes necessary. The review's nuanced content reflects a mixed sentiment that cannot be fully captured by automated systems. For instance, the review praises the product's performance while simultaneously cautioning against certain drawbacks. This blend of sentiments underscores the need for complex context-aware systems or human oversight to reconcile such ambiguities.

Figure 2 presents the comprehensive pseudocode of the algorithm, where:

- *S*: Set of sentiment labels assigned to a review.
- rating: Original star rating of the review (on a scale of 1 to 5).
- counts: Dictionary containing the count of each sentiment label in *S*.
- max_count: Maximum count of a sentiment label in counts.
- has_tie: Boolean indicating if there is a tie among the sentiment labels with the maximum count.
- tied_labels: Set containing the sentiment labels with the maximum count in case of a tie.

Training set preparation introduces an additional step to enhance the quality of the training data. The sentiment label resulting from the majority voting mechanism is compared with the review's original star rating. Both the rating and the sentiment score (ranging from −1 to 1 after normalization) are compared, and reviews with a difference exceeding 1 are excluded. This step aims to filter out potentially unreliable reviews where the content significantly contradicts the user's assigned rating. This could indicate a subjective review, a case of sarcasm, or even a fake review.

In the final stage of our methodology, we utilize the resulting dataset, which has been labeled through our approach, to train a deep learning model for the classification of sentiment in future reviews. The selection of the deep learning architecture is driven by the specific requirements of users and the available computational resources. Among the well-

```
calc_sent(S, rating) → sent :
    counts = count the frequency of each label in S
    max_count = max(counts)
```

$$
\text{has\_tie} = \begin{cases} \text{T,} & \text{if counts contains more than one} \\ & \quad \text{label with count} = \text{max\_count} \\ \text{F,} & \text{otherwise} \end{cases}
$$

```
    if has_tie :
        tied_labels = {l ∈ S | counts[l] = max_count}
        if rating ∈ tied_labels :
            return rating
        else :
            return 'tie'
    else :
        return majority_label(counts)
```

**Figure 2 Majority voting and tie-breaker algorithm.**

established architectures known for their efficacy in sentiment analysis are LSTM networks (*Hochreiter & Schmidhuber, 1997*), which excel at capturing intricate dependencies in text sequences, making them particularly suited for discerning sentiment in complex reviews spanning multiple sentences. RNNs, a broader category encompassing LSTMs and GRUs, are adept at processing sequential data like text, albeit they might encounter challenges with longer sequences compared to LSTMs (*Song, Park & Shin, 2019*). Nevertheless, RNNs offer a simpler architecture that can be more straightforward to implement for sentiment analysis tasks of lesser complexity (*Koutnik et al., 2014*). GRUs, akin to LSTMs, effectively manage long-range dependencies while offering a more streamlined architecture, potentially necessitating fewer training parameters and computational resources (*Chung et al., 2014*). BERT, a modern transformer-based language model, has demonstrated exceptional performance across various NLP tasks, including sentiment analysis (*Devlin et al., 2019*). Its pre-trained embeddings enable effective feature extraction and fine-tuning, facilitating the development of robust and domain-adapted sentiment classifiers.

Any NLP tool inherently carries biases due to differences in training data, algorithms, and sentiment scoring methodologies. Some tools may perform better in specific domains (*e.g.*, social media, product reviews) due to the nature of their training datasets. By combining outputs from multiple tools, the majority voting mechanism dilutes the domain-specific biases of any single tool. Lexicon-based tools may struggle with polysemy or context-specific interpretations of words (*Yang et al., 2020*). The inclusion of machine learning-based tools in our ensemble helps counterbalance these weaknesses. Tools with fixed thresholds for sentiment classification (*e.g.*, neutral *vs*. positive) may result in inconsistent outputs, particularly for mixed or ambiguous cases. Majority voting helps smooth these inconsistencies by aggregating opinions across tools.

By considering factors like review length, desired accuracy, dataset size, and available computational resources, the most suitable architecture can be selected for training the sentiment analysis model. In the context of this research, accuracy refers to the degree to which the majority voting mechanism correctly predicts the sentiment label when compared against the aggregated sentiment derived from the ratings and content of the customer reviews. Complementing accuracy, the reliability is defined as the consistency of sentiment classifications across the multiple tools included in the majority voting mechanism. Our aim is to demonstrate that using the described methodology, one can obtain superior performance, both in terms of accuracy and reliability.

## EXPERIMENTS AND RESULTS

This section describes the data used in our experiments, presents the scenarios taken in consideration and discusses the results obtained.

### Data

The dataset used in our research contains a large collection of reviews (>350 k reviews) and associated metadata from the Flipkart e-commerce platform, collected during 2022–2023 (*Vaghani, 2023*). The data serves as a rich resource for academic research, particularly in the domain of sentiment analysis (*Paul et al., 2017*; *Adane et al., 2023*; *Kanakamedala, Singh & Talasani, 2023*). The review texts contain user-generated content expressing their opinions, experiences, and sentiments regarding the purchased products. The text may vary in length and linguistic style, providing a diverse *corpus* for sentiment analysis. In addition to the review texts, the dataset includes metadata associated with each review, such as product category, brand, price. Typically, each review is accompanied by a numerical rating provided by the user (*e.g.*, star ratings ranging from 1 to 5). Researchers and practitioners may intuitively utilize these ratings as ground truth labels for sentiment analysis tasks. For instance, reviews with higher ratings (4 or 5 stars) may be labeled as positive sentiment, while those with lower ratings (1 or 2 stars) may be labeled as negative sentiment. Reviews with intermediate ratings can be considered neutral or may require further analysis.

The dataset is highly imbalanced (see Fig. 3), the majority of the reviews being five-star labeled and only about 15% of the reviews were negative. This is typical for the online product reviews, as noted since the early days of e-commerce (*Pang & Lee, 2005*; *Mudambi, Schuff & Zhang, 2014*). However, how we will show, the alignment between ratings and sentiment from reviews is, by far, imperfect. Several explanations can be given for the skewness of rating. Users buying a product tend to value it, since they bought it, in the first place. But the review they are providing might contain advice for other users (as a way to give back to the community) that not necessarily represents all-positive sentiments. Therefore, five-stars reviews might also include negative comments about the functionality of an appliance, for example. The following example, extracted from our data, has a five-star rating, but the text suggests the user notes to be "*disappointed to know that I can't use for phone calls*". In this case the rating shows the customer is very happy with the purchase

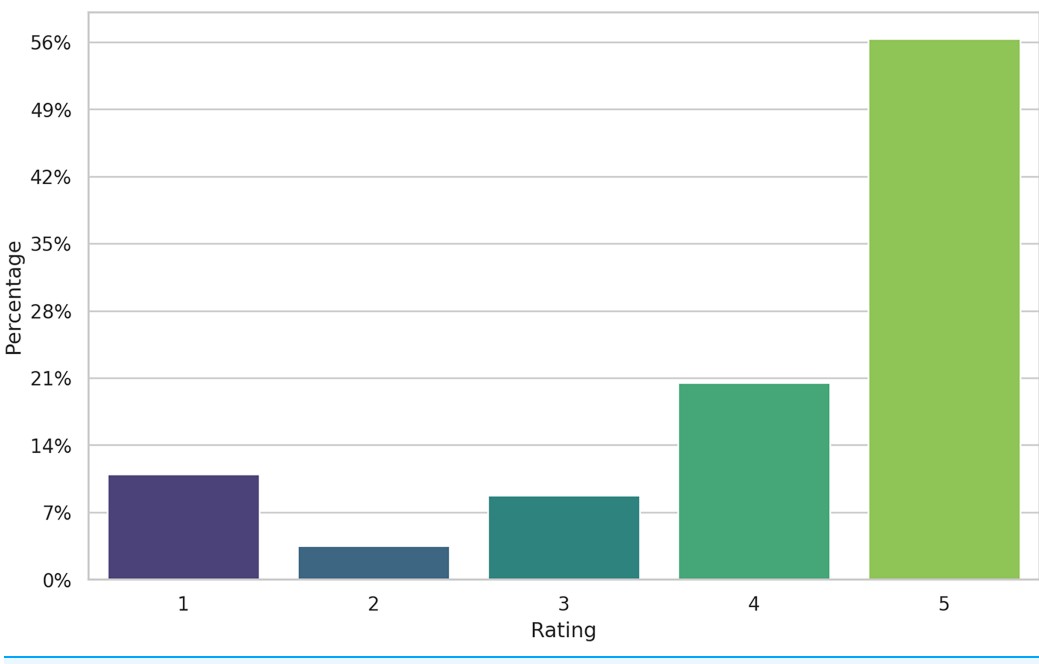

**Figure 3 Distribution of rating in the dataset.**

but chooses to show a missing functionality of the product. This is a typical example when labeling the review as having a positive sentiment based on the rating would be a mistake, only to confuse the training algorithm and ultimately decreasing the performance of the model.

In other cases, such a difference between rating and sentiment within the review can be the cause of a mistake of the user when selecting the rating or even an intentionally misaligned review.

Especially the positive reviews, sometimes comprised of only a few words, can also represent a sign of fraudulent activity, being unuseful for the potential buyer, but boosting the average rating of the product (*Wang & Chen, 2020*).

## Experimental setup

We built our experiments according to the methodology described above (see Fig. 1) taking into account the practical implications of running automatic sentiment classifiers. One of the goals of the research is to assess how much the robustness of the final model varies with the size of the training dataset labeled with the help of the majority voting mechanism. While the cost for using an automatic sentiment classifier for a large dataset can be quite high, the perspective of obtaining good enough results by labeling a small dataset for training a sentiment analysis system looks appealing. For this reason we built three subsets, detailed in Table 2, to test how they will influence the quality of the results. The computing infrastructure used for our experiments is described in Table 3, knowing that the exact configuration details of the cloud platforms used by the four commercial tools are not publicly disclosed.

**Table 2 Datasets roles.**

| Dataset size (# of rows) | Role | Sentiment analysis methods applied |
|---|---|---|
| 21 K | Majority voting labeling | VADER, GOOG, AMZN, IBM, MSFT |
| 50 K | Majority voting labeling | VADER, GOOG, AMZN, IBM |
| 100 K | Majority voting labeling | VADER, GOOG, AMZN, IBM |
| 250 K | Testing | Majority voting trained algorithms |

Note:
Symbols in the table: VADER, Nltk's Vader; GOOG, Google Cloud Natural Language's Analyze Sentiment; AMZN, AWS Amazon Comprehend; IBM, IBM Watson Natural Language Understand, MSFT, Microsoft Azure Text Analytics.

**Table 3 Computing infrastructure for sentiment analysis tools.**

| Sentiment analysis tool | Computing infrastructure |
|---|---|
| Google Cloud Natural Language API | Google Cloud platform |
| Amazon Comprehend | Amazon Web Services (AWS) |
| IBM Watson Natural Language Understanding | IBM Cloud |
| Azure AI Text Analytics | Microsoft Azure |
| VADER, RNN, LSTM, GRU, BERT | Google Colab TPU v2 backend with a 4-chip v2 TPU |

### Extracting sentiment before labeling

When performing sentiment analysis, the preprocessing of text data plays an important role in enhancing the accuracy and reliability of sentiment classification algorithms. When employing lexicon-based approaches, *VADER* in our case, which rely on predefined sentiment lexicons, text preprocessing becomes essential to ensure that the input data aligns with the lexicon's structure and semantics. Preprocessing techniques such as tokenization, stop word removal, and lemmatization help standardize the text input, making it more compatible with the sentiment lexicons used by VADER. By cleaning the text data, noise and irrelevant information are reduced, leading to more accurate sentiment polarity scores. Therefore, we performed preprocessing on our three training datasets, before applying the Vader sentiment classifier.

Conversely, automatic sentiment classifiers provided by commercial tools like Google Cloud Natural Language API, Amazon Comprehend, IBM Watson Natural Language Understanding, and Azure AI Text Analytics often incorporate machine learning models trained on large datasets. These models are designed to handle raw text inputs without the need for extensive preprocessing. Unlike lexicon-based approaches, which rely on predefined word sentiments, machine learning models can learn complex patterns and nuances directly from the data. Therefore, the preprocessing steps necessary for lexicon-based approaches were not required when using these automatic sentiment classifiers.

To replicate the sentiment labeling process using these tools, the following steps were undertaken:

1) Data submission: Reviews were submitted to the APIs using credentials provided by each platform. Authentication details were prepared, and input data was formatted

according to the respective API requirements. Reviews were sent in batches to ensure compliance with API rate limits.

2) Sentiment extraction: Sentiment labels or scores were returned by the APIs in various formats, which are detailed later in the article.

3) Standardization: The outputs from the different APIs were standardized for uniformity to align with the label format used in this study: "Positive," "Neutral/Mixed," and "Negative."

4) Error-handling mechanisms were implemented to manage API request failures and timeouts. The final sentiment labels were stored in a structured CSV format for subsequent analysis.

To elaborate on the sentiment extraction mechanism and the standardization of API responses, we provide additional details on how the outputs from each labeling tool were processed and utilized. VADER provides sentiment analysis results in the form of positive, negative, neutral, and compound sentiment scores for each text analyzed. Positive and negative scores indicate the intensity of respective sentiments, while the neutral score reflects the level of neutrality. The compound score, ranging from −1 to 1, combines all three scores to provide an overall sentiment score for the text, facilitating nuanced understanding of sentiment polarity and intensity. The compound score will be the result used by us in the majority voting process. While most of the automatic classifiers used provide the result as a label, we converted the score to a positive/neutral-mixed/negative label. The literature provides different approaches for this conversion, showing a wide interval used in practice for thresholds, hence the need for experimentation based on data. For example, *Borg & Boldt (2020)* are using the score to build a five-class sentiment labeling for customer reviews: Very Negative (−1 to −0.65), Negative (−0.65 to −0.35), Neutral (−0.35 to 0.35), Positive (0.35 to 0.65), and Very Positive (0.65 to 1). Another work analyzing social media posts, from *Moutidis & Williams (2020)*, uses a three-class labeling, with thresholds for neutral sentiment between −0.05 and 0.05, while scores under −0.05 being considered Negative and those above 0.05, as being Positive, which was also used in the initial release of the Vader model (*Hutto & Gilbert, 2014*). For an application looking for very polarized sentiments in political debates, a thresholds of +/−0.8 was used (*Ramteke et al., 2016*). In our experiments we used the 0.35 as the lower limit of the interval for classifying reviews as positive, and −0.05 as the upper value for the negative sentiment, everything in between the two values being categorized as neutral/mixed. The asymmetrical boundaries (still in the literature recommended limits) would help in better capturing the negative reviews in the context of the highly positive imbalanced dataset.

The Google Natural Language API's sentiment analysis also goes beyond basic labels. Like Vader, it provides sentiment scores between −1.0 (negative) and +1.0 (positive) for each sentence within a text snippet. This offers a fine-grained analysis of sentiment. However, the API also calculates a magnitude score reflecting emotional intensity. This score is particularly useful for longer texts, as it can be skewed by document length.

For short product reviews, the sentiment score alone suffices. Their concise nature often results in low magnitude values regardless of emotional content. Therefore, focusing on the sentiment score provides a sufficient measure of sentiment in this context.

When provided with text input, Amazon Comprehend analyzes the content and assigns sentiment labels along with corresponding confidence scores. The sentiment labels include "Positive", "Negative", "Neutral", or "Mixed", indicating the overall sentiment expressed in the text. For practical reasons of comparability, in this study, we combined the "Neutral" and "Mixed" sentiment labels provided by Amazon Comprehend. This decision was made to streamline the majority voting process and facilitate easier comparison with other sentiment analysis results.

Results are provided in the same way by Azure AI Text Analytics, which employs a combination of machine learning models and linguistic rules to analyze text data, identifying sentiment in the text by assessing the overall emotional tone. "Neutral" and "Mixed" results where also combined for this work.

IBM Watson NLU employs machine learning models to understand the sentiment expressed in the text and assigns sentiment labels such as positive, negative, or neutral.

To assess our hypothesis that significant disagreements exist between the different sentiment analysis tools, we employed Krippendorff's alpha ($\alpha$) as an evaluation metric. $\alpha$ is a statistical measure of inter-rater reliability that quantifies the agreement among multiple raters for a set of items. It accounts for missing data and supports various levels of measurement, such as nominal, ordinal, and interval data. It evaluates how much the observed agreement deviates from what would be expected by chance, based on the following formula:

$$\alpha = 1 - \frac{D_o}{D_e}$$

where: $D_o$ is the observed disagreement among raters, and $D_e$ is the expected disagreement under random assignment.

In general, as suggested by *Krippendorff (2018)*, $\alpha$ values are interpreted as follows: $\alpha \geq 0.80$ indicates strong reliability suitable for most purposes, $0.67 \leq \alpha < 0.80$ reflects moderate agreement, sufficient for exploratory research, and $\alpha < 0.67$ suggests low reliability, raising concerns about the consistency of the ratings, and unreliable for drawing triangulated conclusions. In our case, we calculated $\alpha = 0.73$ indicating a moderate level of agreement among the sentiment analysis tools, suggesting partial consistency in their sentiment classifications but also highlighting notable discrepancies that may arise from differences in algorithmic interpretation or text processing approaches.

### Majority voting and tie-breaker

We ran the algorithms described in the previous subsection on the datasets according to the split in Table 2. Ideally, all algorithms chosen to be part of the majority voting process should be applied to the data and their results subsequently compared according to the decision algorithm. Because of the high costs, we had to limit our experiments by excluding the Azure AI Text Analytics from classifying the 50 and 100 k datasets. Again, for comparing our model's results on the 250 k dataset we had to exclude the IBM Watson, too. After collecting and normalizing all the sentiment labels provided by the algorithms, we applied the majority voting mechanism. The tie breaker rule was used only if a decision
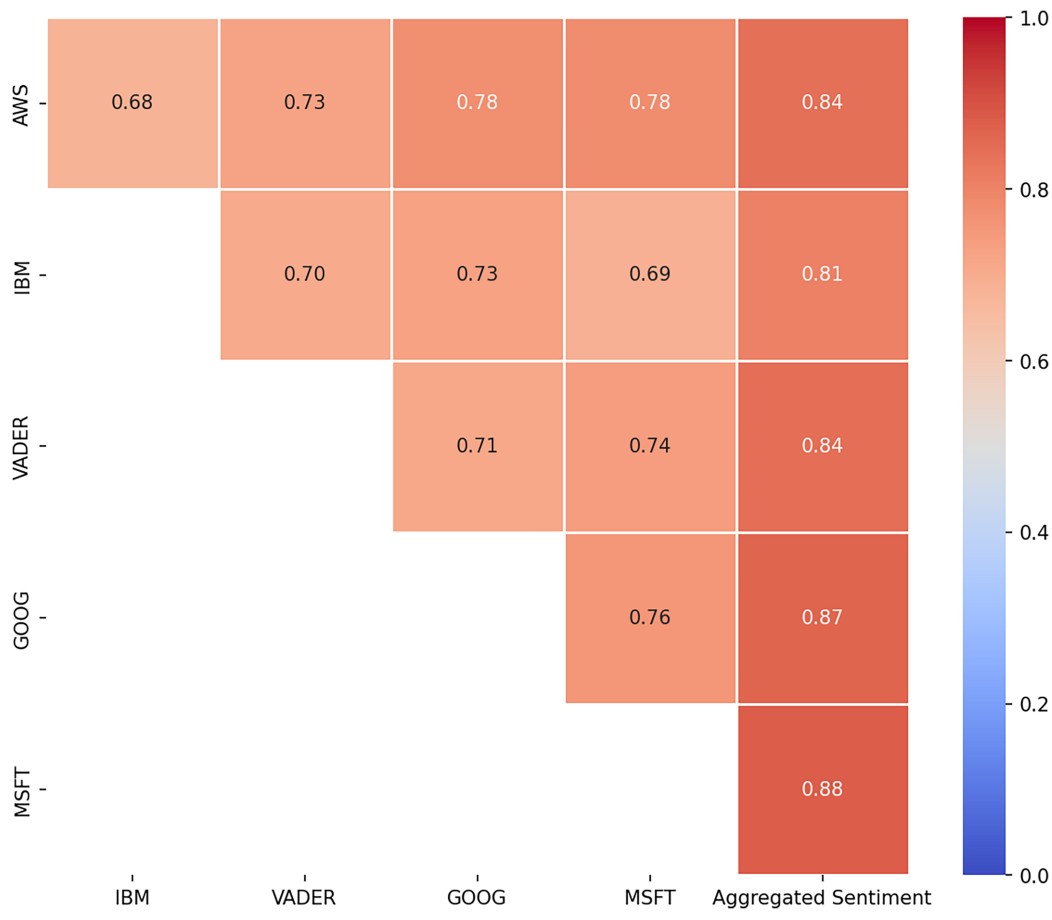

**Figure 4 Krippendorff's alpha after majority voting (Tool-to-Tool and Tool-to-Sentiment Comparisons).**

could not be reached. In some cases, even the use of the normalized rating of the reviews did not help, as explained in the methodology section. Those cases were marked as a "Tie" and excluded from the dataset.

We calculated $\alpha$ again, to assess the agreement between each tool and between each tool and the aggregated sentiment. Naturally, the value of $\alpha$ between the tools and the aggregated sentiment was higher (Fig. 4). It can be observed that the tools from Google and Microsoft had the greatest influence on the aggregated sentiment, as indicated by the highest $\alpha$ values when compared to the individual tools.

### Training the sentiment analysis model

Our models will be built using four deep learning architectures. The selection of RNN, GRU, LSTM, and BERT networks for training our sentiment analysis model reflects their suitability for capturing sequential dependencies and contextual information in textual data, as shown by the literature review. Unlike classical machine learning (ML) methods, which often struggle to effectively model the temporal dynamics and long-range dependencies inherent in sequential data such as text, deep learning architectures excel in

processing and understanding such data (*Colón-Ruiz & Segura-Bedmar, 2020*; *Alantari et al., 2022*).

### Testing the model on new data

Upon completion of training utilizing the deep learning framework, we proceeded to deploy the resultant sentiment analysis model on the dataset comprising 250 k reviews. In pursuit of methodological rigor and ensuring comprehensive evaluation, we subjected the dataset to additional sentiment analysis by employing three of the well-established external tools: VADER, Google Natural Language, and Amazon Comprehend. This comparative approach facilitated a thorough examination of the model's efficacy in sentiment classification, offering insights into its performance vis-à-vis existing state-of-the-art sentiment analysis methodologies.

## Results

This section presents the results obtained after passing the data through the proposed framework and discusses the findings based on our initial goals.

### Applying majority voting to extract sentiment

After running the sentiment analysis methods on the datasets, we were interested in observing the differences between the five methods in labeling the reviews. Figure 5 shows the distribution of the three labels for all the algorithms. The comparative analysis of sentiment distribution across the sentiment analysis tools reveals notable variations in their classification patterns. Interestingly, while all tools exhibit a predominant classification of positive sentiments, which maps the high values of the ratings, there are differences in the percentages of negative and neutral/mixed sentiments.

For instance, VADER exhibits the lowest percentage of negative sentiments (9.60%) compared to other tools, suggesting a more conservative approach in labeling reviews as negative, despite setting the negative upper threshold to −0.05. Conversely, the other algorithms display a higher percentage of negative sentiments (12–13%), indicating a relatively stricter classification threshold for negativity.

In terms of neutral/mixed sentiments, VADER and IBM demonstrate the highest percentages (14%), while GOOG shows the lowest (9.46%). This discrepancy suggests differences in how these tools interpret and classify texts with ambiguous sentiment expressions.

To further explore the data, we constructed a correlation matrix (Fig. 6) to examine the relationship between the sentiment analysis results from each tool and the ratings assigned to the reviews. The ratings, initially ranging from 1 to 5, were rescaled to a uniform scale of −1 to 1, where −1 represents the lowest rating and 1 represents the highest. Similarly, the sentiments extracted by each tool were normalized to the same scale, with −1 denoting negative sentiment, 0 representing neutral or mixed sentiment, and one indicating positive sentiment. By analyzing the correlations between these variables, we aimed to uncover any associations or dependencies between the sentiment expressed in the reviews and the corresponding ratings assigned by the reviewers.

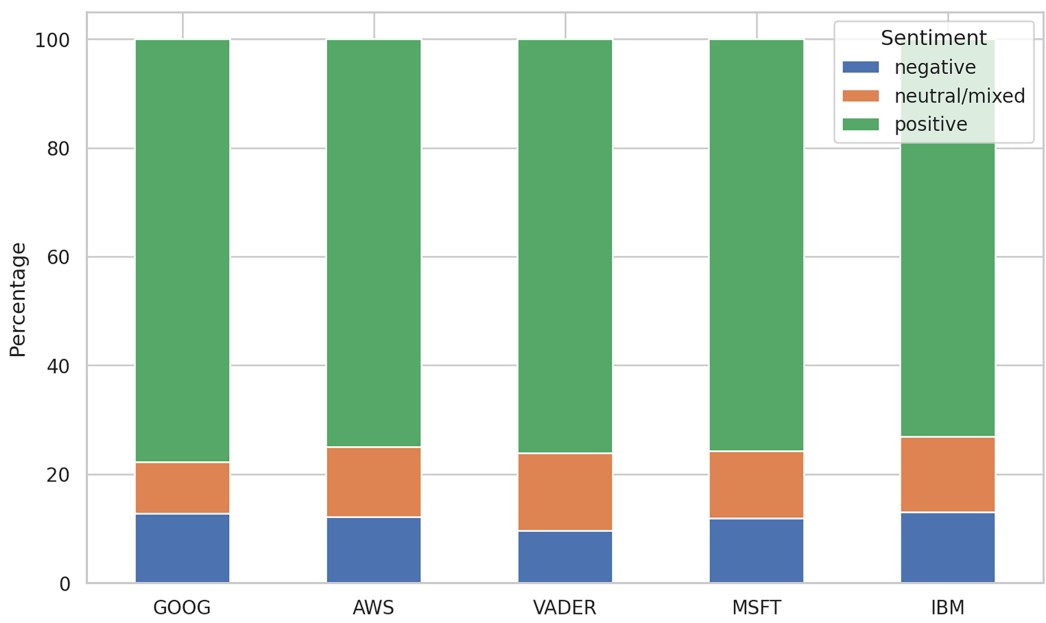

**Figure 5 Distribution of sentiment labels.** MSFT shows the results for training on the 21K review dataset only. IBM doesn't include the results for the 250 k review dataset.

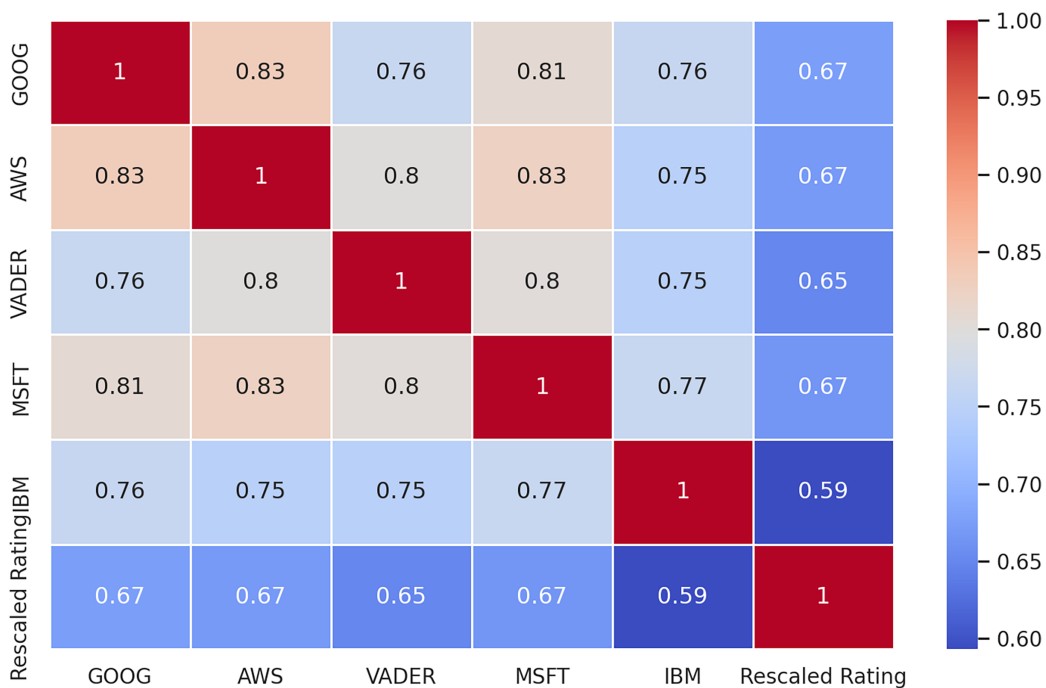

**Figure 6 Correlation matrix between sentiments and ratings.**

The correlation matrix unveils several insights. Firstly, the ratings assigned to the reviews exhibit a moderate to strong correlation, ranging from 0.62 to 0.68, with the sentiment labels derived from the sentiment analysis methods. This suggests a notable, but

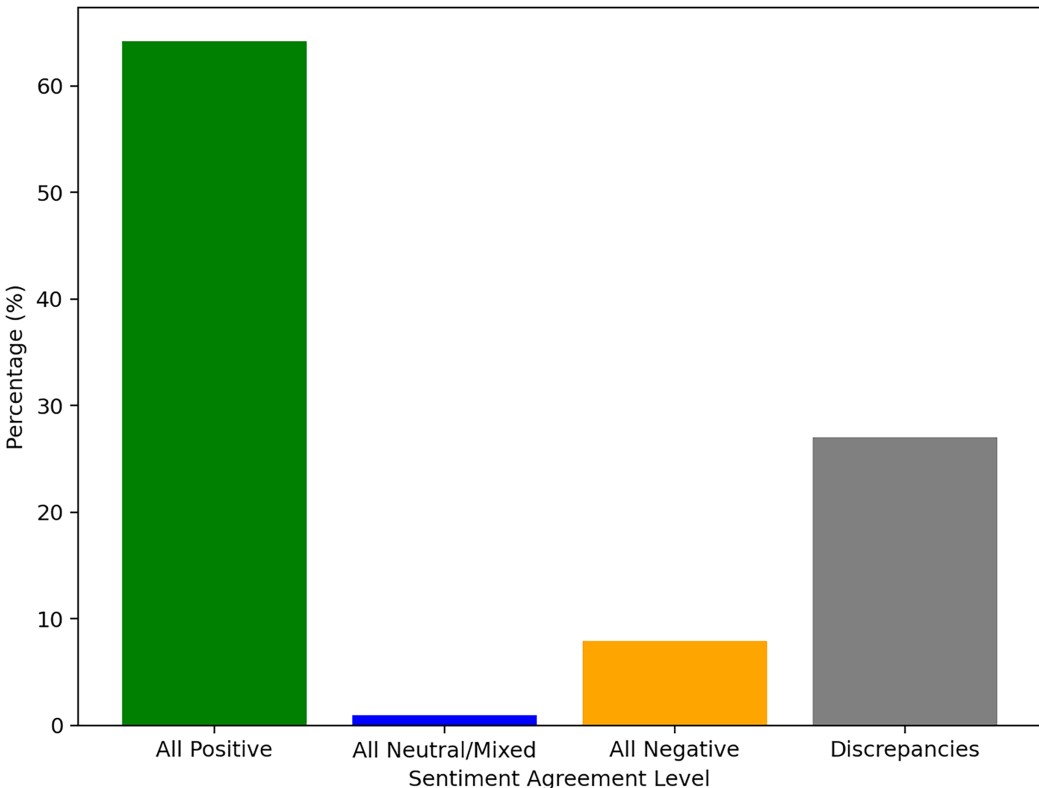

**Figure 7 Agreement and discrepancies between the five reference tools for sentiment analysis.**

far from perfect association between the overall sentiment expressed in the reviews and the numerical ratings provided by the reviewers.

Secondly, the correlation coefficients between the sentiment analysis methods vary, indicating differences in their assessments of sentiment. The highest correlations are observed between Amazon (AMZN) and Google (GOOG) tools, as well as between AMZN and Microsoft (MSFT), both at 0.83. Conversely, IBM exhibits the lowest overall correlation with the other methods, with correlations ranging from 0.75 to 0.77. This suggests that IBM's sentiment analysis outcomes may differ more significantly from those of the other tools.

Measuring the level of agreement between the five reference tools used, we found that about 27% of the reviews in the 21 K review dataset were labeled with a degree of discrepancy between these methods. Figure 7 also shows the difficulties in labeling the neutral/mixed sentiment. Only for a very small proportion all the tools managed to all agree in this matter.

These distinctions are further discernible in Fig. 8, where we depict the ratings associated with each sentiment label across the five methods. While a broad correspondence is apparent between positive sentiments and high ratings, and negative sentiments and low ratings, noteworthy variations among the methods are also evident. Particularly, IBM exhibited the highest proportion of five-star ratings classified as neutral/mixed reviews.

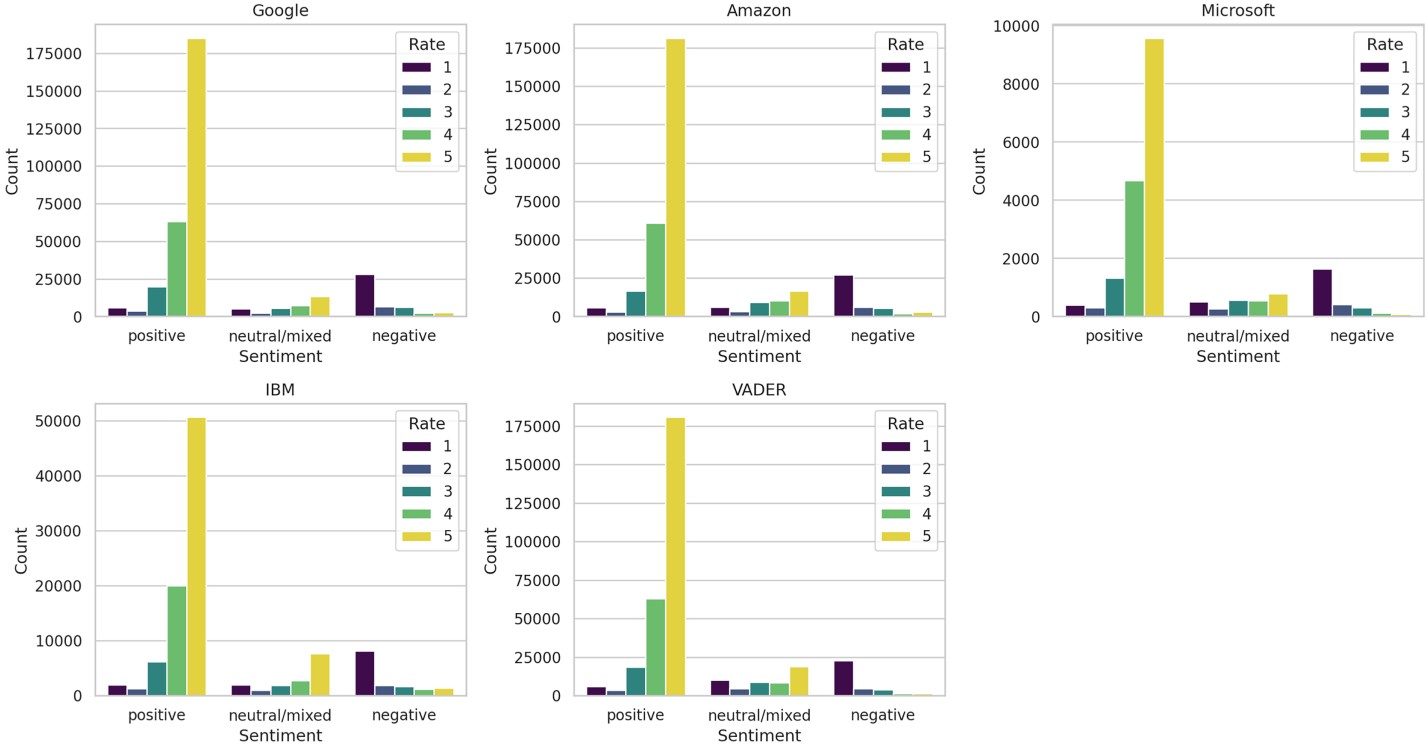

**Figure 8  Distribution of ratings by sentiment label.**

We further aimed to reduce the ambiguities of different decision by different method, labeling the reviews according to the majority voting mechanism. We applied the method on the 21, 50, and 100 k reviews datasets. The results in Table 4 show that the aggregated sentiment is highly correlated with all of the participating methods and also, has the highest correlation among all methods with the rating. We can observe the correlation coefficients for the 50 and 100 k reviews are capped, not improving anymore with the size of the dataset.

We refined the training datasets further, by removing the reviews labeled as a tie. In a real world scenario, these reviews could be manually labeled by a human decider. Next, we verified the difference between the aggregated sentiment and the rating. To exclude from the training data the cases where a large disagreement between the two indicators occured, we set a threshold of one for exclusion. In other words, if there is a disagreement of more than one step between the rescaled rating and the sentiment, the review is eliminated from the training data.

Figure 9 shows a chart with the differences plotted after processing the 100 K reviews dataset.

### Training the deep learning models

We trained our models using the four deep learning architectures (RNN, GRU, LSTM, BERT), using a five-fold process, for each dataset (we also experimented training using

**Table 4  Correlation between aggregated sentiment, methods in majority voting, and rating.**

| Dataset | GOOG | AMZN | VADER | MSFT | IBM | Rating |
|---|---|---|---|---|---|---|
| 21 K | 0.89 | 0.88 | 0.87 | 0.91 | 0.86 | 0.7 |
| 50 K | 0.92 | 0.92 | 0.89 | N/A | 0.87 | 0.7 |
| 100 K | 0.92 | 0.92 | 0.89 | N/A | 0.87 | 0.7 |

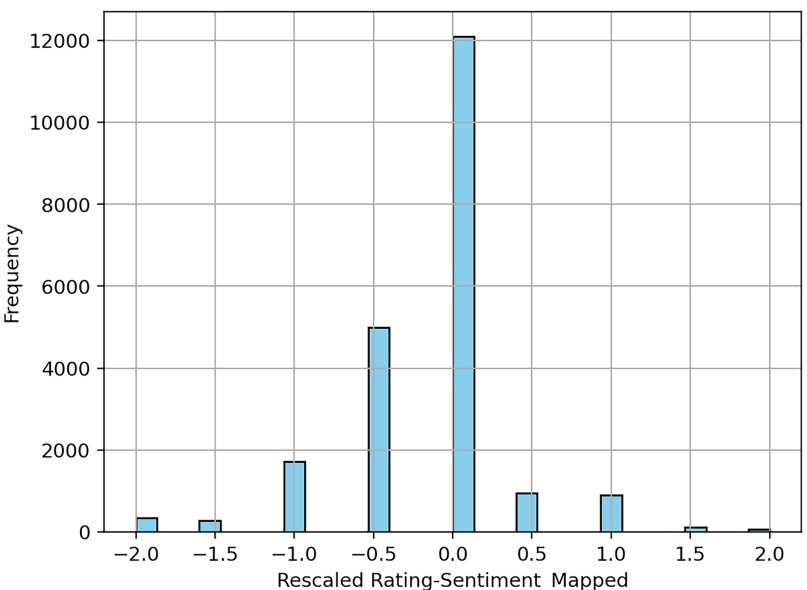

**Figure 9  Differences between sentiment and rating, expressed on a scale from −2 to 2 (*e.g.*, if a sentiment is positive, and the rescaled rating is −1, it would incur a difference of −2).**

10-fold crossvalidation, obtaining minor differences in accuracy, and decided to use the five-fold setup to keep computational costs at a reasonable level). We have chosen Precision, Recall, F1-score, and Accuracy to assess the performance of the deep learning architectures in classifying sentiment as positive, negative, and neutral/mixed for several reasons. Accuracy provides a straightforward measure of overall correctness, indicating the proportion of correctly classified instances among the total instances. However, accuracy alone can be misleading, especially in cases of imbalanced datasets (as ours) where one class may dominate. To address this, we incorporate Precision, which measures the proportion of true positive predictions among all positive predictions, and Recall, which assesses the proportion of true positive predictions among all actual positives. These metrics provide a more nuanced understanding of the model's performance by highlighting the balance between correctly identified positive instances and the total number of instances predicted as positive. The F1-score offers a single metric that balances these two aspects, making it particularly useful when dealing with class imbalances and ensuring that neither Precision nor Recall is disproportionately emphasized.

**Table 5 Validation results for deep learning models on the 21 K review dataset.**

| Method | Accuracy | Std. dev. | Class | Precision | Recall | F1-score |
|---|---|---|---|---|---|---|
| RNN | 0.946 | 0.0017 | Positive | 0.970 | 0.981 | 0.975 |
| | | | Neutral/Mixed | 0.688 | 0.621 | 0.652 |
| | | | Negative | 0.930 | 0.914 | 0.922 |
| GRU | 0.950 | 0.0011 | Positive | 0.971 | 0.983 | 0.977 |
| | | | Neutral/Mixed | 0.710 | 0.627 | 0.665 |
| | | | Negative | 0.935 | 0.924 | 0.930 |
| LSTM | 0.945 | 0.0007 | Positive | 0.971 | 0.976 | 0.974 |
| | | | Neutral/Mixed | 0.676 | 0.621 | 0.646 |
| | | | Negative | 0.925 | 0.936 | 0.931 |
| BERT | 0.964 | 0.0027 | Positive | 0.975 | 0.993 | 0.984 |
| | | | Neutral/Mixed | 0.834 | 0.679 | 0.748 |
| | | | Negative | 0.955 | 0.946 | 0.951 |

Across all methods, the models achieved highest levels of accuracy on the 110 K review dataset, with BERT obtaining 97.5%, RNN achieving 95.7%, GRU achieving 95.8%, and LSTM achieving 95.8% (comparing Tables 5–7). The differences, compared with the other two datasets, are not large, especially compared to the 50 K dataset. However, a critical examination reveals potential areas for improvement, particularly in the classification of the neutral/mixed sentiment class. This is more obvious on the results from the 21 K review dataset, with precision, recall and F-1 score under 0.7, with the exception of BERT (Table 5). These values were improved by adding more training data (Tables 6 and 7). This indicates a challenge in accurately capturing instances of neutral or mixed sentiment, potentially leading to misclassifications or inconsistencies in sentiment analysis outcomes. BERT outperformed the other architectures, with more significant differences in accurately predicting neutral/mixed reviews, although it required more computational resources to train, which is consistent with other research reviewed (*Tao & Fang, 2020*; *Talaat, 2023*). Also, the obtained accuracy was ≈ 2% higher compared to other studies using the same dataset (*Kanakamedala, Singh & Talasani, 2023*; *Aishwarya Bharathy & Princy Suganthi Bai, 2024*). To assess the robustness of the findings, we calculated the standard deviation of the accuracy across the five folds. The low standard deviations indicate good stability in the models' performance across different data partitions, underscoring the reliability of the proposed methodology.

### Applying the trained models on new data

We used the remaining portion of the Flipkart dataset (250 k reviews) to test our models. Without access to the ground truth labels for the reviews, we considered the best option for evaluating the performance of our models would be to compare their classifications with the results provided by Google, Amazon, and Vader tools. Due to high computational costs, the tools from Microsoft and IBM were omitted in this step.

**Table 6 Validation results for deep learning models on the 50 K review dataset.**

| Method | Accuracy | Std. dev. | Class | Precision | Recall | F1-score |
|---|---|---|---|---|---|---|
| RNN | 0.953 | 0.0010 | Positive | 0.975 | 0.983 | 0.979 |
| | | | Neutral/Mixed | 0.744 | 0.729 | 0.736 |
| | | | Negative | 0.934 | 0.888 | 0.911 |
| GRU | 0.954 | 0.0001 | Positive | 0.973 | 0.986 | 0.979 |
| | | | Neutral/Mixed | 0.764 | 0.720 | 0.741 |
| | | | Negative | 0.941 | 0.890 | 0.914 |
| LSTM | 0.953 | 0.0001 | Positive | 0.974 | 0.984 | 0.979 |
| | | | Neutral/Mixed | 0.750 | 0.721 | 0.736 |
| | | | Negative | 0.938 | 0.889 | 0.913 |
| BERT | 0.964 | 0.0057 | Positive | 0.990 | 0.980 | 0.985 |
| | | | Neutral/Mixed | 0.843 | 0.750 | 0.794 |
| | | | Negative | 0.944 | 0.930 | 0.937 |

**Table 7 Validation results for deep learning models on the 110 K review dataset.**

| Method | Accuracy | Std. dev. | Class | Precision | Recall | F1-score |
|---|---|---|---|---|---|---|
| RNN | 0.957 | 0.0006 | Positive | 0.981 | 0.982 | 0.982 |
| | | | Neutral/Mixed | 0.751 | 0.773 | 0.762 |
| | | | Negative | 0.930 | 0.907 | 0.919 |
| GRU | 0.958 | 0.0002 | Positive | 0.978 | 0.985 | 0.981 |
| | | | Neutral/Mixed | 0.774 | 0.751 | 0.763 |
| | | | Negative | 0.930 | 0.907 | 0.918 |
| LSTM | 0.958 | 0.0009 | Positive | 0.979 | 0.984 | 0.982 |
| | | | Neutral/Mixed | 0.766 | 0.754 | 0.760 |
| | | | Negative | 0.931 | 0.909 | 0.920 |
| BERT | 0.975 | 0.0001 | Positive | 0.989 | 0.990 | 0.989 |
| | | | Neutral/Mixed | 0.854 | 0.846 | 0.850 |
| | | | Negative | 0.954 | 0.951 | 0.952 |

In Fig. 10 we show the results obtained, by plotting the sentiments *vs* rating. In this way we can observe the differences among the various models. The results indicate that the trained models are strong candidates for sentiment extraction tasks. Among these models, those employing deep learning methodologies with the majority voting approach appear to better capture positive sentiments correlated with high ratings. However, they tend to slightly under-perform compared to Google and Amazon tools in capturing negative sentiments associated with low ratings, and showing a tendency to classify more reviews with a rating of two as neutral/mixed. Furthermore, while the majority voting trained models associate more three-star reviews as neutral/mixed compared to the Google tool, they exhibit a tendency towards classifying them as positive in a larger proportion than their piers.

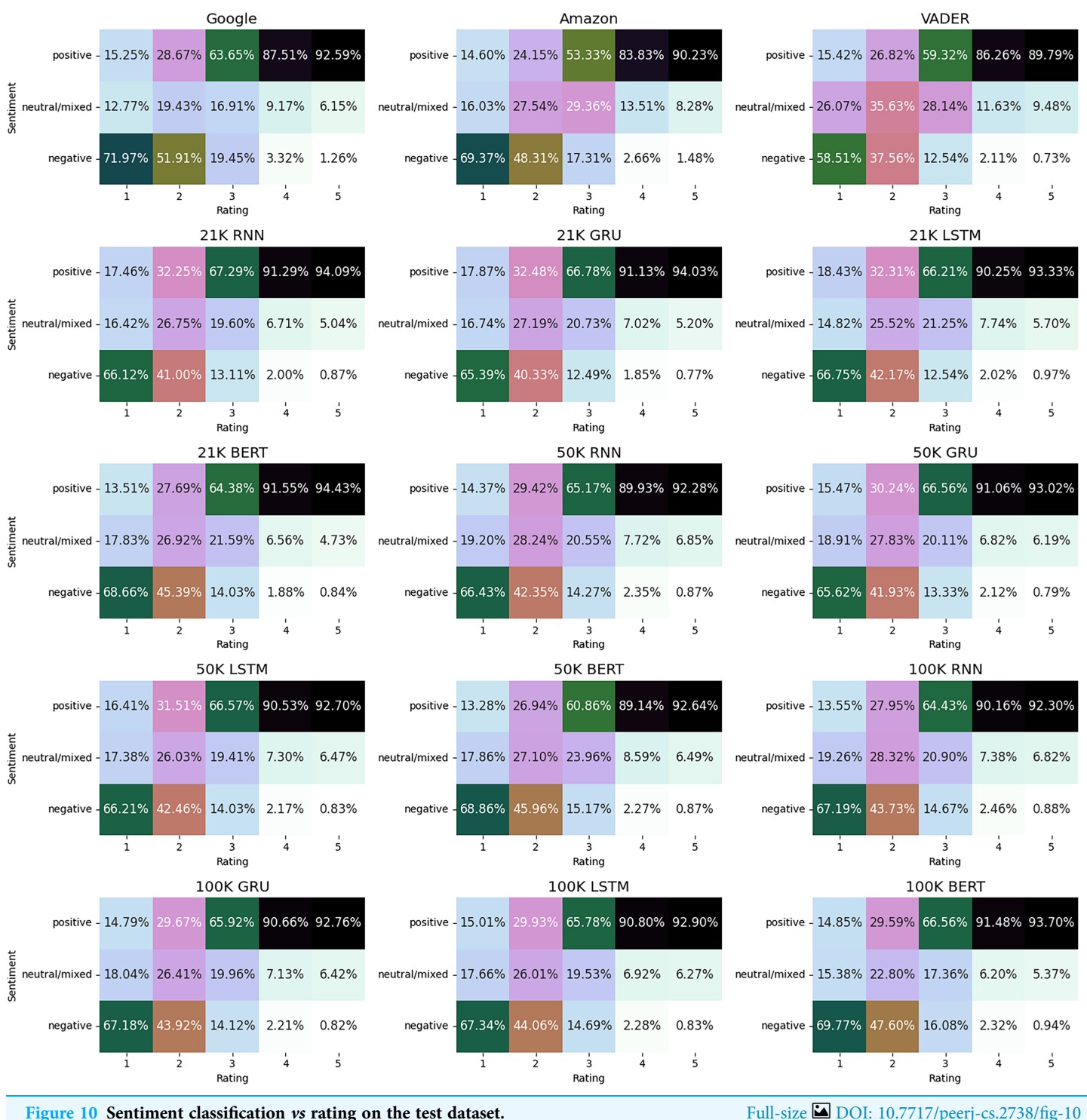

**Figure 10 Sentiment classification *vs* rating on the test dataset.**

The larger the training dataset, the closer the resulting models become. All three deep learning architectures appear to converge towards the same decisions, as evidenced by the correlation matrix depicted in Fig. 11. Furthermore, increasing the training dataset from

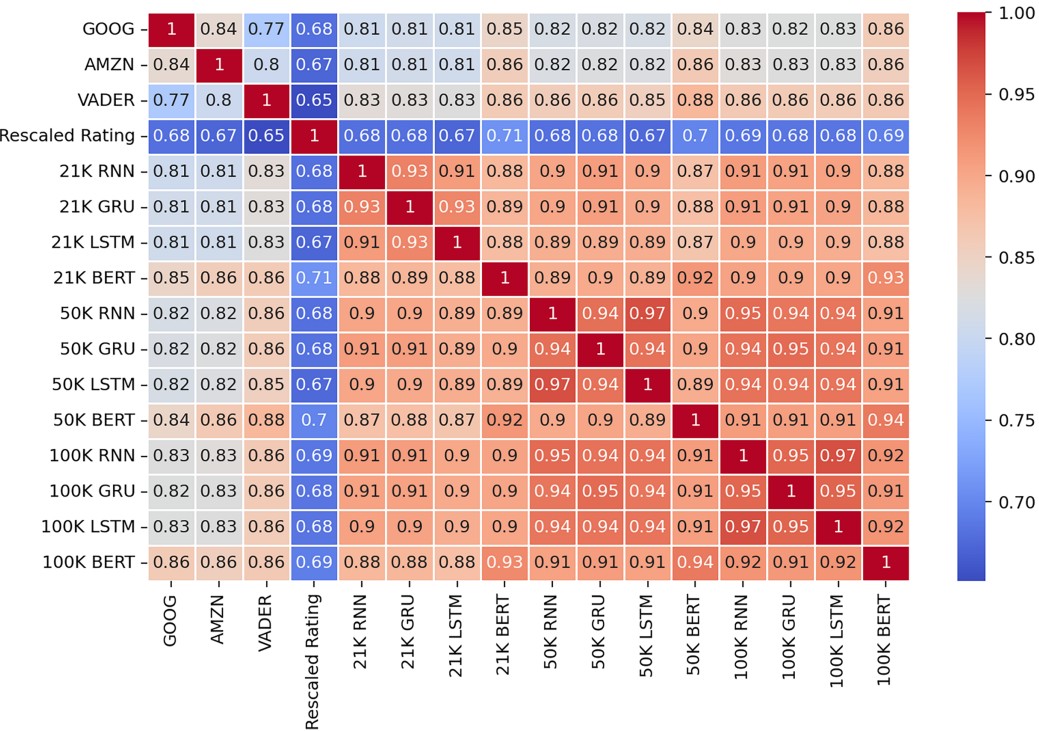

**Figure 11 Correlation matrix of the results from the test dataset.**

50 K reviews to 100 K reviews did not significantly alter the level of correlation between the trained models and the reference models used in the majority voting mechanism. The new models exhibit a strong positive correlation with the reference models and a slightly lower correlation with the ratings of the reviews. Notably, this latter correlation remains approximately at the same levels as that shown by the reference models, thereby demonstrating the robustness of our proposed method.

## CONCLUSION

Our work aimed to introduce a novel methodology for creating trustworthy sentiment labels for product reviews by implementing a majority voting system. This system aggregates sentiment labels from various methods and utilizes review ratings as a tiebreaker. The enhancement in labeling quality is achieved prior to the training phase, leading to more accurate and reliable sentiment analysis results. The results demonstrate that, even with relatively small datasets, deep learning architectures can be trained to achieve performance comparable to widely recognized systems. The consistency of the results underscores the robustness of the majority voting mechanism in mitigating labeling ambiguities and improving the overall trustworthiness of sentiment analysis outcomes.

However, despite this promising outcome, some disparities between predicted sentiment and review ratings persist. Both our models and the reference methods encounter difficulties, particularly when dealing with neutral/mixed reviews, also noted by *Sazzed & Jayarathna (2021)* in a research with similar goals. Further investigation into the

factors contributing to these discrepancies, such as the diversity and complexity of language expressions conveying neutral or mixed sentiment, is warranted. Addressing these challenges could enhance the models' ability to accurately discern nuanced sentiment nuances, thereby improving the overall reliability and effectiveness of sentiment analysis in real-world applications.

One potential method for enhancing classification accuracy could involve further improving the quality of the training data. This could be achieved by incorporating manually labeled reviews that were excluded as ties by the majority voting and tie-breaker method. By enriching the training data with high-quality annotations, we may mitigate the challenges associated with ambiguous sentiment labeling and further improve the performance of sentiment analysis models.

While the majority voting mechanism offers a promising approach to mitigating the challenges of ambiguous sentiment labeling, its optimal performance remains an open question. *Biswas, Young & Griffith (2022)* evaluated the efficiency of an automated labeling system, finding it to be approximately 80% accurate compared to human labeling. While we believe that the diversity of sentiment analysis tools and a consensus-based majority voting approach can significantly enhance the accuracy and reliability of sentiment analysis, further benchmarking research is necessary to definitively establish its superiority.

We also acknowledge the need for real-world case studies to further validate the practical applicability and robustness of the proposed approach. To address this, future research could include applications of the method to datasets from diverse domains, such as social media sentiment analysis, customer service logs, and product reviews from different industries. Despite the higher costs associated with training an ensemble model like ours, the long-term savings could be substantial due to the benefits of a more reliable model compared to the automated tools used in its construction.

## ACKNOWLEDGEMENTS

During the preparation of this work the author used Chat-GPT in order to improve language. After using this tool/service, the author reviewed and edited the content as needed and take full responsibility for the content of the publication.

### Funding

This work was funded by the 2023 Development Fund of Babeș-Bolyai University. The funders had no role in study design, data collection and analysis, decision to publish, or preparation of the manuscript.

### Grant Disclosures

The following grant information was disclosed by the authors:
2023 Development Fund of Babeș-Bolyai University.

## Competing Interests

The author declares that they have no competing interests.

## Author Contributions

- Darie Moldovan conceived and designed the experiments, performed the experiments, analyzed the data, performed the computation work, prepared figures and/or tables, authored or reviewed drafts of the article, and approved the final draft.

## Data Availability

The data is available at Kaggle:

Nirali Vaghani. (2023). Flipkart Products Review Dataset (363K Data) [Data set]. Kaggle. https://doi.org/10.34740/KAGGLE/DSV/5051290.

The code and curated data are available at Github and Zenodo:

- https://github.com/dariemoldovan/majority-voting-reviews-sentiment.

- dariemoldovan. (2025). dariemoldovan/majority-voting-reviews-sentiment: Majority Voting Sentiment for Reviews (v1.0). Zenodo. https://doi.org/10.5281/zenodo.14855834.

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
