# Peer review of "A majority voting framework for reliable sentiment analysis of product reviews"

_PeerJ Computer Science, doi:10.7717/peerj-cs.2738_

## Round 0.1 · original submission · Major Revisions

Dear authors,

Thank you for submitting your article. Feedback from the reviewers is now available. It is not recommended that your article be published in its current format. However, we strongly recommend that you address the issues raised by the reviewers, especially those related to readability, experimental design and validity, and resubmit your paper after making the necessary changes.

Best wishes,

Reviewer 1 ·

Basic reporting

When referring to previous research, in Introduction and Related Works the author tends to treat the references disjointed. The focus should be on creating a narrative that shows the need for the current research (i.e., establishing a niche). The RW section ends with a generic paragraph: “The literature review showed some common difficulties in extracting sentiment analysis from product reviews, mainly related to the quality of the input data. Trustworthy mechanisms for evaluating the concordance of the e-commerce rating systems and the content of the review are needed to ensure the quality of a sentiment classifier. While the technology evolution has improved the processing capacity of the big data, there are still steps to be made to ensure highly relevant inputs.”
The author might benefit from using John Swales’s Creating a Research Space [C.A.R.S.] Model.

Experimental design

The article is within the aims and scope of the journal and the methods are satisfactorily described. Still, many limitations are listed:
• "We used the remaining portion of the Flipkart dataset (250k reviews) to test our models. Without access to the ground truth labels for the reviews, we considered the best option for evaluating the performance of our models would be to compare their classifications with the results provided by Google, Amazon, and Vader tools. Due to high computational costs, the tools from Microsoft and IBM were omitted in this step."
• "Because of the high costs, we had to limit our experiments by excluding the Azure AI Text Analytics from classifying the 50k and 100k datasets. Again, for comparing our model’s results on the 250k dataset we had to exclude the IBM Watson, too"
The lack of ground truth labels seems to be a problem as there is a lack of independent benchmarking to truly asses the performance. Furthermore, isn’t comparing the results from an ensemble method (the majority voting mechanism) with the results of individual sentiment analysis tools that were part of that ensemble problematic? There seems to be some circular reasoning going on.

Validity of the findings

Majority voting is largely used in ensemble methods where different methods are used together. Furthermore, algorithms such as random forests or bagging use similar approaches internally. So, I wouldn’t quite say that “This paper introduces a novel majority voting method …” as it is stated in the Abstract.
The author provides the code which encourages replication. A problem is that the method requires access to several services such as pre-trained automatic sentiment classifiers: Google Cloud Natural Language API, Amazon Comprehend, IBM Watson Natural, Language Understanding, and Azure AI Text Analytics. The CSVs already have these scores: AWS, GNLP, IBM, MSFT, VSC, Rescaled Rating, Sentiment difference, VADER, GOOG, Sentiment, Sentiment_Mapped. So, the provided code does not contain the full approach.
There is no Discussion section. The results are narrated in a generic, LLM-like way. For example:
“The correlation matrix unveils several insights. Firstly, the ratings assigned to the reviews exhibit a moderate to strong correlation, ranging from 0.62 to 0.68, with the sentiment labels derived from the sentiment analysis methods. This suggests a notable, but far from perfect association between the overall sentiment expressed in the reviews and the numerical ratings provided by the reviewers.”
The author should elaborate on the implication of the findings, contextualize them within the broader field, and address potential limitations. As it is, the approach seems overly complicated, and it is not very clear as if it has real-world applications potential, including cost-wise.

Additional comments

No other comment.

Cite this review as

·

Basic reporting

The introduction of the paper provides a solid background on sentiment analysis and outlines the motivation for the study. It successfully explains the challenges in classifying sentiment due to discrepancies in user reviews and ratings. However, the introduction could benefit from a clearer statement about the specific research gap the paper addresses, as well as a stronger articulation of why this majority voting method is necessary in comparison to existing approaches.
The technical terms are well explained, especially the description of the majority voting mechanism and how various sentiment analysis tools are combined. The methodology is thorough, but it could be more effective if key terms like "accuracy" and "reliability" were defined with greater clarity, particularly in the context of statistical performance. Additionally, a more formal treatment of the results—perhaps including a discussion of statistical significance or robustness—would strengthen the findings. The structure of the paper aligns well with PeerJ standards, but there are areas where conciseness could be improved, especially in the literature review. The results are well supported by experimental data, though a deeper discussion of why certain methods perform better in specific contexts could enhance the impact of the findings.

Experimental design

When using Flipkart data set to test model performance, since Flipkart data set does not have labels that need to be generated by Google and other tools, but these methods are themselves used in the model, is it inappropriate;
You should elaborate more on the preprocessing steps for each dataset used in the experiments. For example, how were stop words handled across different sentiment classifiers, and was there any normalization of the review text?
Your reasons for choosing the sentiment classifier need to be clearer and explain why it was specifically chosen for the majority voting system to improve.

Validity of the findings

It is feasible to adopt the majority voting mechanism, but there is no ablation experiment to prove that its efficiency is optimal;
You provide comprehensive experiments with different data set sizes, but seem to rely too heavily on automated sentiment analysis tools, with less discussion of the biases these tools can introduce and how they affect study results;
You effectively solve the problem of data imbalance by adopting a majority voting mechanism, but there is a lack of real case studies to demonstrate the practical applicability and robustness of the proposed method in different data set;
Your analysis needs to be more intuitive. I suggest you improve the presentation of lines 187-191 to make the conclusion more clear and credible

Additional comments

The layout format can be improved, for example, there is a large amount of white space on page 10;
The algorithm diagram given on page 5 has some ambiguities, and there is room for improvement;
sentiment analysis models used in this article are all classic models, such as LSTM. Is it possible to use a newer model to judge the performance of the model, for example:
BERT: Pre-training of Deep Bidirectional Transformers for Language Understanding
A novel cascade model for end-to-end aspect-based social comment sentiment analysis
Sentiment Knowledge Enhanced Pre-training for Sentiment Analysis;

Reviewer 3 ·

Basic reporting

1. Introduction
• On line 24 & 25, please support the statement with more references. There is a break in logical sequence from line 26 to the next paragraph which started on line 27. Reference (Fang and Zhan, 2015) on line 30 is an old reference to support this claim. Please look for new reference and it should be multiple claims (references). Please don’t make statements you cannot support with literature. For instance, line 45 - 49, no reference. Are you speaking from your mind or based on literature. Line 50 – 56, no reference, on what bases are you making those claims. On line 57, you discussed what your research addresses. Please without reference to literature, you cannot address any issue. You cannot create your own literature gaps based on what you think and address it. The introduction has to be written again with statements and paragraphs supported by literature. The introduction should also explain sentiment analysis and discuss its relevance to businesses over the years. This should be detailed and backed with literature.
2. Related Work
• Line 84. Please is that a sub-heading or what? And it starts with On?
• Please you need to discuss textual emotions in sentiment analysis, sentiment analysis using supervised, unsupervised learning algorithms. Before you can discuss deep learning for sentiment analysis
• The literature review should have a summary in the form of a table with natural processing technique, feature selection mechanism, dataset category, algorithms used, best classifier and main shortfall.
• We need a thorough review to ascertain the relevance of your study.

Experimental design

3. Methodology
• Please the referenced diagram Figure 1, is it novel? Did you modify any diagram. Please clarify and reference if any.
• Please you need to explain your methodology into details. What is tokenization, stop word removal, stemming etc. Even a lay person should understand your work.
• Again from line 210 to 219, you made a claim of well-established architectures known for their efficacy in sentiment analysis. That is why a detailed literature review and summarized table is necessary to reveal this claim. This claim for now is baseless.
• You need to discuss into details the algorithms you implemented. The machine learning metrics to ascertain performance. You need to discuss those.
4. Experiment and Results
• Good reference to data source and justification from line 226 to 237
• Good gap identified in data from line 238 to 250
• The experiment and results section was written properly

Validity of the findings

5. Results
• Why did you use the 5-fold cross validation technique. You can compare it to the 10-fold
• Aside that, the results section was thoroughly discussed.
• But the findings should be referenced back to literature for discussion.
6. Conclusion
• Good conclusion with gaps identified for further research.

Additional comments

This is a good paper that need some modifications.

Cite this review as

---

## Round 0.2 · Minor Revisions

Dear Authors,

Thank you for the revised paper. It is requested that minor revisions be made in accordance with the recommendations provided by the referees, and that the manuscript be resubmitted accordingly.

Warm regards,

Reviewer 1 ·

Basic reporting

Overall, I find the explanations satisfactory.
The authors deleted the last sentence from the Related Works section, but they didn’t replace it with content that clearly establishes the niche and motivation of their paper. How does this article contribute to existing research (including the articles mentioned in Table 1) that also evaluate the concordance between e-commerce rating systems and the content of the reviews?

Experimental design

“The provided code demonstrates the majority voting methodology using pre-calculated sentiment scores, allowing replication without paid API access. However, as the original reviews data are freely available, users can independently run the tools to replicate all experiments.”
Still, some guidance could be provided on how to replicate this preprocessing part. The reviewers' guideline asks: "Is there a discussion on data preprocessing and is it sufficient/required?"

Validity of the findings

Seems fine!

Cite this review as

·

Basic reporting

The methodology section of the paper is described clearly and in detail, particularly with thorough explanations of the design of the majority voting mechanism and the combination of sentiment analysis tools. Although the authors introduced Krippendorff's alpha to measure consistency, they did not clearly define key terms such as "accuracy" and "reliability." This may lead to ambiguity in readers' understanding of these terms, affecting the comprehensive interpretation of the research results. It is recommended to provide clear definitions of these key terms in the methodology section, especially within the context of statistical performance, to enhance the rigor and clarity of the article. Additionally, a more formal treatment of the results, including discussions on statistical significance or robustness, would further strengthen the credibility and persuasiveness of the research findings.

Experimental design

The paper constructs a sentiment analysis framework based on a majority voting mechanism, selecting multiple automated sentiment analysis tools and integrating their results to improve the consistency and reliability of sentiment labels. The experimental design is rigorous, and the methodology is described in sufficient detail to allow other researchers to replicate the experiments.
However, although the authors mention that the selection of these tools is based on their diversity and complementarity, the explanation of the specific advantages and limitations of each tool remains relatively brief. It is recommended to provide a clearer explanation of the reasons for choosing specific sentiment classifiers and to elaborate on the specific strengths and weaknesses of each tool in sentiment analysis. This would help readers better understand the rationale behind the tool selection and the comprehensiveness of the experimental design.

Validity of the findings

The paper validates the effectiveness of the majority voting mechanism through experiments and demonstrates its advantages in reducing data labeling ambiguity and improving model performance. The experimental results show that sentiment labels generated using the majority voting mechanism significantly enhance the accuracy of deep learning models, particularly in handling neutral/mixed sentiments.
Although the authors explain that the design of the majority voting mechanism aims to reduce biases from automated sentiment analysis tools, they do not provide a detailed discussion of each potential bias or explain how the majority voting mechanism mitigates these biases. Comparisons of experimental results could be used to support these discussions.

Additional comments

In future research, it would be beneficial to consider using more diverse datasets, including reviews in different languages, from different platforms, and across various product categories, to validate the model's generalization capabilities. Additionally, integrating a user feedback mechanism, allowing users to provide feedback and corrections on sentiment classification results, could further enhance the model's accuracy.

---

## Round 0.3 · accepted · Accept

Dear Author,

I would like to express my gratitude for the lucid manner in which you have addressed the comments made by the reviewers. It is evident that the paper has undergone significant improvement and is now in a state that is suitable for publication.

Best wishes,

Reviewer 1 ·

Basic reporting

Seems fine!

Experimental design

Seems fine!

Validity of the findings

Seems fine!

Cite this review as